# Antisolvent controls the shape and size of anisotropic lead halide perovskite nanocrystals

Kilian Frank [1,5], Nina A. Henke [2,5], Carola Lampe[2], Tizian Lorenzen[3], Benjamin März [3], Xiao Sun [4], Sylvio Haas [4], Olof Gutowski[4], Ann-Christin Dippel [4], Veronika Mayer[2], Knut Müller-Caspary [3], Alexander S. Urban [2] ✉ & Bert Nickel [1] ✉

Colloidal lead halide perovskite nanocrystals have potential for lighting applications due to their optical properties. Precise control of the nanocrystal dimensions and composition is a prerequisite for establishing practical applications. However, the rapid nature of their synthesis precludes a detailed understanding of the synthetic pathways, thereby limiting the optimisation. Here, we deduce the formation mechanisms of anisotropic lead halide perovskite nanocrystals, 1D nanorods and 2D nanoplatelets, by combining in situ X-ray scattering and photoluminescence spectroscopy. In both cases, emissive prolate nanoclusters form when the two precursor solutions are mixed. The ensuing antisolvent addition induces the divergent anisotropy: The intermediate nanoclusters are driven into a dense hexagonal mesophase, fusing to form nanorods. Contrastingly, nanoplatelets grow freely dispersed from dissolving nanoclusters, stacking subsequently in lamellar superstructures. Shape and size control of the nanocrystals are determined primarily by the antisolvent's dipole moment and Hansen hydrogen bonding parameter. Exploiting the interplay of antisolvent and organic ligands could enable more complex nanocrystal geometries in the future.

Semiconductor nanocrystals possess unique size-dependent properties for their use in widespread applications, such as light-emitting diodes, solar cells, photocatalysis, and biomedical imaging[1–4]. Lead halide perovskites (LHPs) are an emerging class of material with advantageous characteristics, such as strong optical absorption, high photoluminescence (PL) quantum yields (QYs), and optical responses tunable throughout the visible range via size and halide composition[5–7]. Moreover, LHPs exhibit a high propensity for forming low-dimensional nanocrystals, such as zero-dimensional (0D)

quantum dots[8], one-dimensional (1D) nanorods and nanowires[9–12], and two-dimensional (2D) nanoplatelets and nanosheets[13–16]. These 1D and 2D nanocrystals show favourable anisotropic properties, such as directional and polarised light emission and superior emission in the blue spectral range[6,7,17–19]. Previous studies have focused on developing methods to produce anisotropic LHP nanocrystals and have shown various synthetic parameters influencing size and geometry[20–22]. Obtaining the high yield and high homogeneity essential for commercialisation or the use in hierarchically assembled materials[23,24]

[1]Soft Condensed Matter Group and Center for NanoScience, Faculty of Physics, Ludwig-Maximilians-Universität München, Geschwister-Scholl-Platz 1, Munich, Germany. [2]Nanospectroscopy Group and Center for NanoScience, Faculty of Physics, Ludwig-Maximilians-Universität München, Königinstraße 10, Munich, Germany. [3]Department of Chemistry and Center for NanoScience, Ludwig-Maximilians-Universität München, Butenandtstraße 11, Munich, Germany. [4]Deutsches Elektronen-Synchrotron DESY, Notkestraße 85, Hamburg, Germany. [5]These authors contributed equally: Kilian Frank, Nina A. Henke. ✉e-mail: urban@lmu.de; nickel@lmu.de

requires fine-tuning the synthesis and a complete understanding of the synthetic process, also in view of potential upscaling. The most simple liquid-phase synthesis of colloidal nanocrystals progresses via a three-stage process of supersaturation, nucleation, and growth, as given by the classical LaMer model[25]. Here, the nucleation and growth phases must be separated to control nanocrystal formation. Furthermore, a timely termination of the synthesis is essential before Ostwald ripening leads to nanocrystal size defocusing.

Using modern analytical tools, this timing has been realised for metal nanoparticles[26] and conventional semiconductor nanocrystals[27,28], whose syntheses often take tens of minutes to hours and which can be easily separated into nucleation and growth steps. However, LHP nanocrystal syntheses are typically finalised within seconds[14,25], severely impeding real-time analysis and, thus, a tailored anisotropic nanocrystal fabrication. A substantial slowing down of the growth of LHP nanocrystals has been realised by controlling the availability of halide[22,29], and, recently, by replacing the organic ligands present during synthesis[8,30]. This allowed for a detailed analysis of the growth mechanism, yet it often changed the final shape of the resulting nanocrystals. Since many synthesis studies focus mainly on the final product, nucleation, growth, and potential intermediate phases are only vaguely understood[31]. For example, it is widely believed that intermediate clusters, micelles, or complexes play a decisive role in regulating reaction kinetics by binding reactants, which are slowly released as monomers[8,16,32,33] however their dimensions are rarely reported unambiguously[34]. Moreover, the labile ligand binding in LHPs potentially promotes nonclassical growth mechanisms like oriented attachment, fusion, and recrystallisation of intermediates, yet the exact mechanisms at work are elusive[33,35]. The addition of a so-called antisolvent, i.e., a poor solvent for the as-formed nanocrystals, is assumed to act as a structure-directing agent[36,37] and has shown to be beneficial for fabricating anisotropic nanocrystals[6,38]. However, due to a lack of structural information during the rapid synthesis, a strategy for control of LHP nanocrystal anisotropy and size is still lacking.

In this work, we conduct an in situ structural and spectroscopic study to elucidate the synthesis pathway of anisotropic LHP nanocrystals, and provide a guideline for their tailored fabrication. We use simultaneous small- or wide-angle X-ray scattering (SAXS and WAXS) and PL spectroscopy at a 3rd generation storage ring-based X-ray source. We can decouple the individual steps of nanocrystal formation by simultaneously obtaining structural, crystallographic, and optical data of the synthesis with a temporal resolution down to 50 ms. Accordingly, we observe that prolate and emissive nm-sized CsPbBr₃ nanoclusters form nearly instantaneously upon precursor mixing and act as reaction intermediates for both 1D nanorods and 2D nanoplatelets. Subsequent injection of larger volumes of the antisolvent

acetone at higher Cs to PbBr₂ precursor ratio induces the formation of a dense, hexagonal mesophase of the intermediates, wherein they fuse to form nanorods. In the absence of this mesophase, intermediate nanoclusters and dissolved precursor ions contribute to the growth of freely dispersed nanoplatelets, representing the thermodynamically most stable structure under these conditions[39]. These assemble into lamellar superstructures, a process which potentially enhances size homogeneity. We ascertain that two specific solvent properties are crucial for controlling the shape and monolayer (ML) thickness of anisotropic perovskite nanocrystals, namely the dipole moment $\mu$ and Hansen hydrogen bonding parameter $\delta_H$[40]. We identify a narrow range of solvent conditions with significant promise for obtaining more complex LHP nanocrystal shapes. Exploring this solvent range and the interplay between solvent, organic ligands, and precursor concentrations will likely lead to new syntheses for anisotropic nanocrystals.

## Results and discussion
### Synthesis scheme

Through trial and error experiments and, recently, the use of machine learning, we have developed synthesis strategies to obtain anisotropic CsPbBr₃ nanocrystals with precise control over their dimensions (Fig. 1a)[38,41]. The synthesis is conducted in ambient conditions and can be described as a ligand-assisted spontaneous crystallisation. The synthesis commences with a PbBr₂ precursor in toluene, oleic acid, and oleylamine, into which a second Cs-oleate precursor is injected and thoroughly mixed for 10 s. Subsequently, acetone, a moderately polar antisolvent, is injected into this reaction mixture. After 60 – 120 s, the reaction is terminated by centrifuging and redispersing the precipitate in n-hexane, serving as the final purification. The formation of aniso-tropic nanocrystals is promoted by a stoichiometric deficiency of Cs⁺ ions, while the nanocrystal thickness depends on the ratio of Cs-oleate to PbBr₂ precursor. Slight variations in this ratio and relative acetone volume lead to the formation of either quasi-1D nanorods or quasi-2D nanoplatelets, as depicted in Fig. 1b, c. To understand how the nano-crystal shape is controlled, we first focused on two types of nano-crystals with distinct morphological and optical characteristics: 3ML nanorods and 2ML nanoplatelets. The nanocrystals, with sizes of (1.8 × 1.8 × 15) nm³ and (1.2 × 15 × 15) nm³, respectively, as evidenced by transmission electron microscopy (TEM, Fig. 1c and Supplementary Note 1, Supplementary Figs. 1 and 2), exhibit strongly blueshifted absorbance and PL spectra compared to bulk CsPbBr₃ (Fig. 1d and Supplementary Fig. 3). The PL emission, centred at 460 nm (full width at half maximum (FWHM) = 18 nm) and 434 nm (FWHM = 14 nm), respectively, confirms the formation of nanocrystals with pronounced quantum and dielectric confinement and narrow size distribution[11,13]. Powder X-ray diffraction (PXRD, Supplementary Fig. 4) and PL

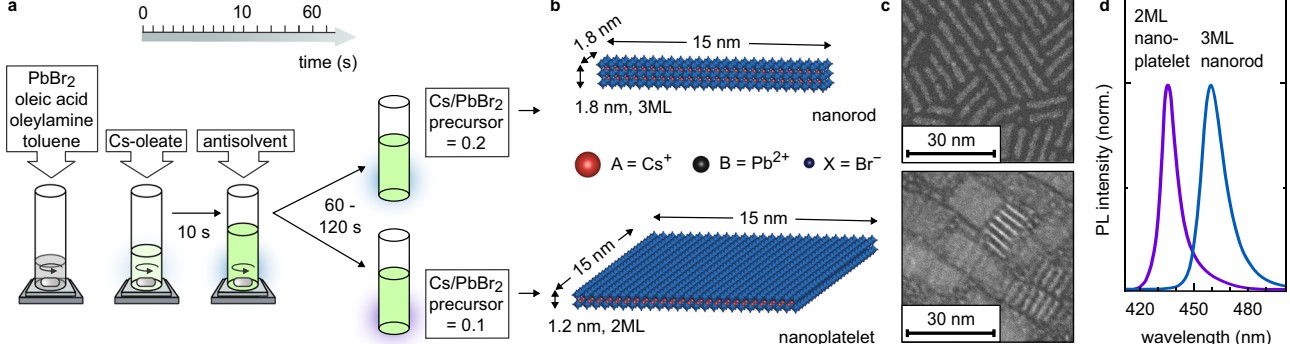

**Fig. 1 | Synthesis and characteristics of the anisotropic perovskite nanocrys-tals. a** Scheme of the room temperature synthesis of CsPbBr₃ nanorods and nanoplatelets. **b** Sketch of individual CsPbBr₃ 3 monolayer (ML) nanorods and 2ML nanoplatelets with typical dimensions. **c** Annular dark field scanning transmission electron microscopy (ADF-STEM) images of 3ML nanorods and 2ML nanoplatelets. **d** Ex situ photoluminescence (PL) spectra of 3ML nanorods and 2ML nanoplatelets. Source data are provided as a Source Data file.

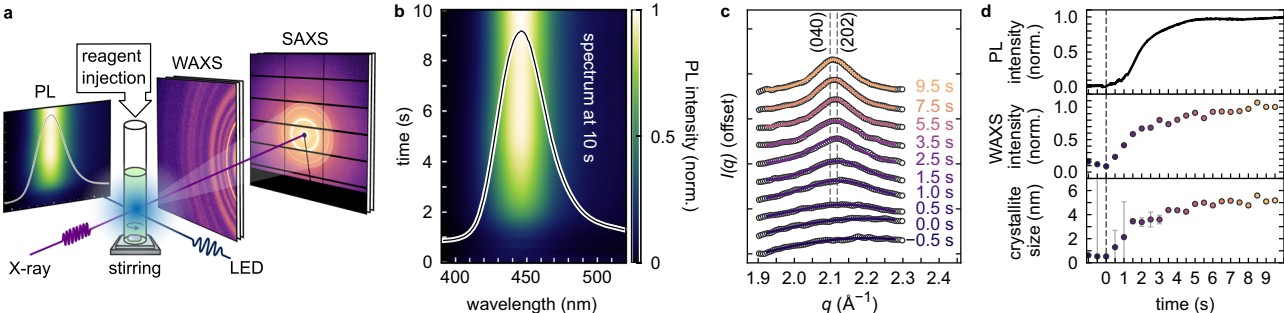

**Fig. 2 | In situ WAXS and PL spectroscopy reveal the formation of nanoclusters during Cs-oleate and PbBr₂ precursor mixing. a** Sketch of the in situ reaction cell for simultaneous small- and wide-angle X-ray scattering (SAXS, WAXS), and photoluminescence (PL) spectroscopy. **b** In situ PL data recorded in between injection of Cs-oleate into the PbBr₂ precursor ($t = 0$ s) and antisolvent addition ($t = 10$ s). The PL maximum position at 446 nm and full width at half maximum (FWHM) =

40 nm do not change over time, while the PL intensity increases rapidly, saturating after 5 s. **c** Simultaneously recorded in situ WAXS data, showing the formation of the (0 4 0) and (2 0 2) CsPbBr₃ Bragg reflections between 0.5 and 2 s after Cs-oleate injection. **d** PL intensity, WAXS peak intensity, and crystallite size as derived from a Scherrer analysis. Error bars correspond to the propagated standard error of the fitted WAXS peak width. Source data are provided as a Source Data file.

excitation spectroscopy (PLE, Supplementary Fig. 5), show no non-emissive Cs₄PbBr₆ byproduct. Since TEM imaging can be ambiguous in distinguishing rod and platelet shape depending on the observed orientations[41,42], we additionally confirm the nanocrystal shape via SAXS in solution (Supplementary Fig. 6).

The question remains, however, how to achieve control of anisotropy and ML thickness based on rational principles. To investigate this, we built a small reaction cell (Supplementary Fig. 7) for integration into the beamlines of the PETRAIII storage ring at DESY, Hamburg, to simultaneously monitor PL and either SAXS or WAXS. This cell allows us to follow structural and optoelectronic properties during synthesis with a time resolution down to 50 ms, as shown in Fig. 2a and detailed in Supplementary Notes 2 and 3. In the experiments, a glass capillary is loaded with PbBr₂ precursor solution (Fig. 1a) and a magnetic stirring bar. Cs-oleate precursor is injected through a motorised syringe pump, starting the reaction at $t = 0$ s, and the antisolvent is injected through a separate motorised syringe pump at $t = 10$ s.

**Intermediate nanocluster formation after precursor mixing**

As soon as Cs-oleate is added to the PbBr₂ precursor, a bright blue emission, located at 446 nm (FWHM = 40 nm), and indicative of strongly quantum-confined CsPbBr₃ nanoclusters, is detected (Fig. 2b)[43–45]. Simultaneously, a growing WAXS signal (Fig. 2c) is observed, with a Bragg reflection emerging at $q = 2.1$ Å⁻¹, i.e., at the position of the (0 4 0) and (2 0 2) reflections of orthorhombic CsPbBr₃ (Supplementary Figs. 8 to 10)[46]. The simultaneous emergence of the PL and WAXS signals indicates that crystalline perovskite forms immediately after precursor mixing. After 5 s, both signal intensities reach a plateau (Fig. 2d), suggesting the initial burst of nucleation is caused by supersaturation in response to the injection of the second precursor and quickly terminated, as previously reported[30]. A Scherrer analysis of the WAXS signal (see "Methods" and Fig. 2d) reveals that nanoclusters reach a crystallite domain size of $(5.1 \pm 0.3)$ nm after 5 s. However, as the PL emission wavelength does not shift during this synthesis phase, shown in Supplementary Fig. 11, at least one nanocluster dimension must maintain its size within the strongly quantum-confined regime. Importantly, these crystallites are still smaller than the final nanocrystals, and their PL spectra do not match those of either nanorods or nanoplatelets (Supplementary Fig. 12). Accordingly, we identify these LHP nanoclusters as reaction intermediates[30].

The rapid formation of nanoclusters and evolution of their physical dimensions were followed in more detail by in situ SAXS measurements (Supplementary Note 3), shown here from slightly before Cs-oleate injection up to the moment before antisolvent injection at

$t = 10$ s (Fig. 3a). After Cs-oleate injection ($t = 0$ s), an immediate increase in SAXS intensity at low $q$ is observed, followed by saturation within a few seconds, in agreement with PL and WAXS data in Fig. 2b–d. A fit to the signal of the PbBr₂ precursor before mixing (solid line at $t = -0.5$ s) shows that the PbBr₂ precursor is dispersed in the form of micellar nanoclusters, with 1.2 nm diameter PbBr₂ cores enclosed in ligand micelles of 2.8 nm diameter (see Supplementary Fig. 13 and Supplementary Table 4). Each micelle contains on average $9.6 \pm 2.2$ PbBr₂ units (Supplementary Table 5). Absorption spectra of the PbBr₂ precursor, shown in Supplementary Fig. 14, further corroborate the micellar structure. The internal structure of the PbBr₂ core was further analysed via the reduced pair distribution function (PDF) obtained from X-ray total scattering (Supplementary Fig. 15 and Supplementary Note 2). The distances between Pb and Br ions in these micelles are in agreement with an octahedral coordination.

We analyse the time-dependent SAXS data with a two-component model form factor, accounting for the micellar cluster structure of the PbBr₂ precursor as the first component and the newly forming LHP nanoclusters as freely dispersed ellipsoids as the second component (Fig. 3b, c, Supplementary Figs. 16 to 18 and Supplementary Tables 6 and 7). This model describes the SAXS data very well, allowing us to quantify the nanocluster formation in terms of size and density. As shown in Fig. 3d, the number density of nanoclusters rapidly increases within 1–2 s. It then slowly decays, indicating a phase of coalescence or Ostwald ripening. The nanoclusters are prolate ellipsoids and their remaining growth proceeds slowly within 10 s from initially $(1.8 \pm 0.2)$ nm × $(4.3 \pm 0.2)$ nm to a final size of $(2.1 \pm 0.2)$ nm × $(5.9 \pm 0.4)$ nm. Importantly, the short axes, which dominate confinement, retain their sizes. This explains the stationary PL maximum observed in Fig. 2b. A comparison of the core sizes of the PbBr₂ precursor micelles and the nanoclusters reveals that initially 4 to 6 PbBr₂ micelles contribute to nucleation, and further material is incorporated during growth (Supplementary Table 5). As a consequence, we observe a decline of the number density of precursor micelles by 25% during nanocluster nucleation, shown in Supplementary Fig. 19. Importantly, freely dissolved precursor ions in the solution are not expected to play a significant role due to their insolubility in toluene. This distinguishes the reaction pathway from, e.g., biphasic syntheses where water can serve as an ion reservoir[47]. In addition to in situ SAXS, we verified the presence of prolate intermediate nanoclusters through ex situ TEM imaging of the crude reaction solution obtained after precursor mixing (Fig. 3f). We observe a monolayer of size-uniform nanoclusters arranged in a regular hexagonal assembly with a spacing of $(5.01 \pm 0.13)$ nm (Fig. 3g and Supplementary Fig. 20). Both synthesis schemes show identical intermediate nanocluster sizes at this stage (Fig. 3e), i.e., the

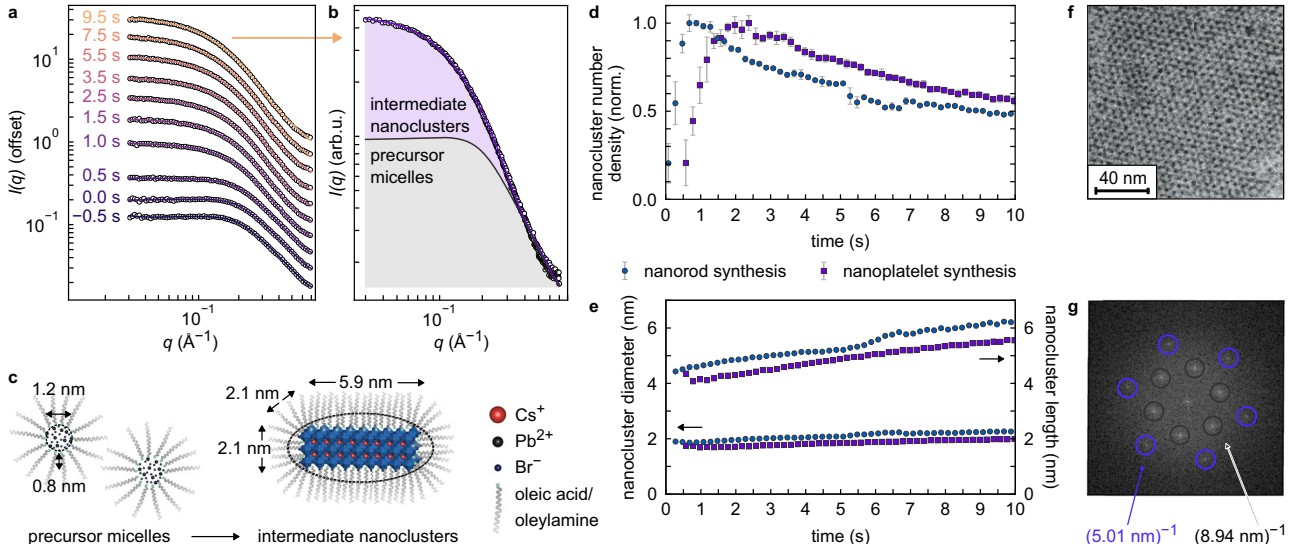

**Fig. 3 | Intermediate nanoclusters are identical in nanorod and nanoplatelet synthesis, according to SAXS and TEM data. a** In situ small-angle X-ray scattering (SAXS) intensity recorded in between injection of Cs-oleate into the $PbBr_2$ precursor ($t = 0$ s) and antisolvent addition ($t = 10$ s). Solid lines indicate fits to the data. **b** Decomposition of the SAXS intensity into signals of precursor micelles and intermediate nanoclusters, illustrated for $t = 9.5$ s, immediately before antisolvent addition. **c** Dimensions of the $PbBr_2$ precursor micelles and intermediate nanoclusters obtained from SAXS data analysis. **d** Number density of nanoclusters as a function of time from SAXS data analysis for nanorod and nanoplatelet synthesis, respectively. Error bars correspond to the standard deviation of four consecutive fit results. **e** Diameter and length of the intermediate nanoclusters from SAXS data analysis. **f** Transmission electron microscopy (TEM) image of a reaction solution drop casted on a substrate after precursor mixing. **g** Fast Fourier transform (FFT) of the TEM image in (**f**) indicates a hexagonal assembly with 5.01 nm and 8.94 nm distance to the nearest and next-nearest nanocluster. Source data are provided as a Source Data file.

syntheses of nanorods and nanoplatelets proceed via the same prolate intermediates.

### Nanorod growth via hexagonal mesophase formation

Up to this point in the synthesis, there is essentially no difference between the 2ML nanoplatelet and 3ML nanorod syntheses. Accordingly, the divergence of the final product shape must be induced by the subsequent antisolvent injection. The moderately polar antisolvent acetone is well-suited to study the mechanism at work. Injection of a larger relative volume of acetone antisolvent results in 3ML nanorods while a smaller relative volume of acetone yields nanoplatelets (see "Methods" and Supplementary Table 8). Indeed, the SAXS signal evolution differs substantially for nanorod and nanoplatelet synthesis. For the nanorod synthesis (Fig. 4a), three intense peaks appear immediately after antisolvent injection. These peaks are located at the positions of $q_{100} = 0.150$ Å⁻¹, $q_{110} = \sqrt{3} \cdot q_{100}$, and $q_{200} = 2 \cdot q_{100}$, corresponding to the diffraction pattern of a two-dimensional hexagonal structure. The hexagonal lattice constant is $d = 4\pi/(\sqrt{3}q_{100}) = (4.8 \pm 0.1)$ nm. The spacing corresponds to the short axis of prolate intermediates (1.8 nm) plus an intercalated double layer of ligands (3.0 nm)[41], suggesting this hexagonal phase is formed by dense packing of the intermediate nanoclusters. Upon close inspection, the SAXS intensity shown in Fig. 4a reveals an additional weak intensity peak between 0.04 and 0.07 Å⁻¹, which is not part of the 2D hexagonal (h k 0) series (Supplementary Fig. 21a). This (0 0 1) peak indicates the formation of order along the long axis of the intermediates as they grow to nanorods, i.e., the intermediates stack end-to-end with a spacing that increases over time and finally reaches $2\pi/q_{001} = (13.7 \pm 0.2)$ nm (Supplementary Fig. 22). Thus, the evolution of the long axis of the nanorods during synthesis from initially 9.8 nm (including ligand shell) up to 13.7 nm can be estimated from the (0 0 1) position. Additional information on the nanorod formation can be obtained from the PL intensity. Within 5 s after acetone injection, PL decreases and then gradually recovers throughout the measurement time (Fig. 4b), with the final PL spectrum being slightly redshifted

compared to the intermediate nanoclusters (Supplementary Fig. 12). The decrease of the PL hints at a destabilisation of the intermediate nanoclusters by the antisolvent, which may facilitate growth into rods.

### Nanoplatelet growth followed by lamellar mesophase formation

In contrast, upon injection of a smaller volume of acetone, as needed for the nanoplatelet synthesis, the emerging X-ray pattern of the assembly differs considerably from the previous two-dimensional hexagonal pattern. Now, a (h 0 0) peak series appears with integer multiples of $q_{100} = 0.143$ Å⁻¹. This signifies the formation of a lamellar phase with repeat distances of $2\pi/q_{100} = (4.4 \pm 0.1)$ nm, indicating stacking of 1.4 nm thick nanoplatelets separated by an intercalated double layer of ligands (3.0 nm). Furthermore, the SAXS intensity shows an additionally weak intensity (0 0 l) peak series starting at smaller $q$-values, indicating the formation of an even larger lattice formed by these stacks (Fig. 4e and Supplementary Fig. 22). The final repeat distance of this (0 0 l) peak series corresponds to $2\pi/q_{001} = (12.1 \pm 0.4)$ nm, implying the presence of nanoplatelets with a uniform lateral size of 12.1 nm, including ligand shell, arranged side by side. The emergence of the lamellar superstructure peaks is delayed by 60 s (Fig. 4e and Supplementary Fig. 21b) with respect to antisolvent injection. This is in contrast to the nanorod synthesis, where the superstructure peaks occur within a few seconds, and suggests that nanoplatelets form while freely dispersed and subsequently stack in a lamellar assembly. This is also supported by the PL data, which show a sharp decrease of the PL centred at 451 nm associated with the intermediate nanoclusters. A blue-shifted PL peak emerges 20 s after antisolvent injection. The emission wavelength of 433 nm is characteristic for 2ML nanoplatelets[13,14,41]. Subsequently, PL emission of the platelets increases, and platelets form stacks, as evidenced by SAXS.

### Effects of antisolvent addition

The two synthesis pathways proceed fundamentally differently after the addition of the same antisolvent in different volumes (Fig. 4c, d, g, h). This can be rationalised as follows: Firstly, the addition of acetone

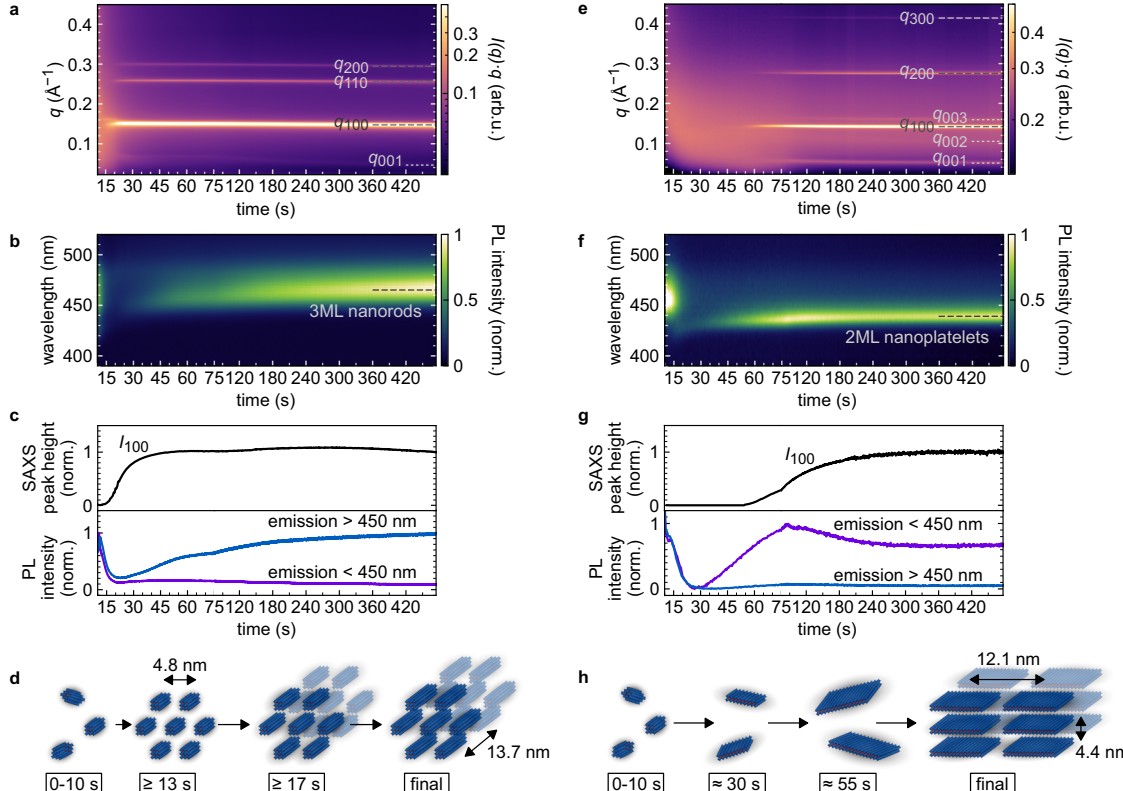

**Fig. 4 | Antisolvent injection induces mesophase formation, enabling fusion of intermediate nanoclusters into nanorods. a** In situ small-angle X-ray scattering (SAXS) intensities of a 3 monolayer (ML) nanorod-forming reaction mixture after antisolvent injection as a function of scattering vector and time. **b** In situ photo-luminescence (PL) intensities of a 3ML nanorod-forming mixture as a function of wavelength and time. The initially broad PL of intermediate nanoclusters is strongly reduced by antisolvent injection. Later, the characteristic emission of 3ML nanorods at > 450 nm is recovered. **c**, SAXS and PL intensities extracted from (**a** and **b**), summarised. **d** Sketch of hexagonal mesophase formation and

transformation of prolate intermediate nanoclusters to nanorods. **e** In situ SAXS intensities of a 2ML nanoplatelet-forming reaction mixture after antisolvent injection. **f** In situ PL intensities of a 2ML nanoplatelet-forming mixture. The initial PL from intermediate nanoclusters is suppressed in favour of a sharp peak at 433 nm, characteristic of the formation of 2ML nanoplatelets. **g** SAXS and PL intensities extracted from (**e** and **f**), summarised. **h** Sketch of the sequence of growth of freely dispersed nanoplatelets from intermediate nanoclusters and subsequent stacking of nanoplatelets. Source data are provided as a Source Data file.

increases the polarity of the reaction mixture and therefore, the solubility of precursor ions in the solution is enhanced. At the same time, the ligand shell of the nanoclusters is disturbed[48], and surface defects are introduced[49]. Both effects reduce the stability of the intermediate nanoclusters, which are partly dissolved, releasing ions into the solvent mixture. This is supported by the observation of a strong reduction in PL intensity of the intermediates (Fig. 4c, g) and their WAXS signal (Supplementary Fig. 9) upon the addition of the antisolvent.

Secondly, as long as the nonpolar ligand shell is still intact, the solubility of LHP nanoclusters as a whole is largely reduced by the addition of polar solvent[50,51]. Thus, the rapid addition of a large volume of acetone results in an instantaneous formation of a hexagonal mesophase formed by densely packed prolate nanoclusters, in essence, a phase separation. The destabilisation of the intermediate nanoclusters follows the mesophase formation temporally, indicating that ion and ligand release takes at least a few seconds while phase separation is almost instantaneous. In turn, the rod-like nanocrystals form within the hexagonal mesophase in the presence of the desta-bilised intermediates, presumably by fusion and recrystallisation, as suggested by increasing PL emission and WAXS intensity (Supplementary Figs. 9 and 11). By varying the precursor concentration as well as the antisolvent volume, we find that an increased relative concentration of intermediate nanoclusters leads to assembly in the mesophase, similar to observations for CdS magic-sized clusters[52], but not to nanorod growth, as the destabilizing effect of the antisolvent is then insufficient (see Supplementary Fig. 23). We also verified that

hexagonal mesophase formation occurs when the LHP nanoclusters are present and does not occur in mixtures containing only ligands or one precursor salt (Supplementary Fig. 24).

At lower acetone volumes and/or precursor concentrations, the effect of enhanced LHP ion solubility dominates. These reaction conditions result in the growth of nanoplatelets. The nanoplatelets grow laterally in size while freely dispersed and only later begin to stack in a lamellar phase due to a combination of solvation forces[53], van der Waals interactions between large platelet facets[54], and attractive dispersion forces between alkyl chains of surface ligands[55]. Nanoplatelet stacking becomes apparent as a set of peaks at regular intervals in the SAXS intensity (Fig. 4e) and a reduction of the PL intensity due to increased reabsorption and scattering (Fig. 4g, Supplementary Fig. 25, Supplementary Table 9). The platelet shape is likely the thermo-dynamically stable polymorph since 2ML nanoplatelets also form very slowly without antisolvent injection (Supplementary Fig. 26). In this case, variation of the precursor concentration moderately affects the reaction rates (Supplementary Fig. 27 and Supplementary Table 10). Thus, we identify two processes contributing to 2ML nanoplatelet growth: direct supply of available precursor material from solution and supply from dissolution of intermediate nanoclusters, which serve as an ion reservoir and regulate the reaction kinetics.

## Shape and size control by tuning solvent properties
So far, our analysis highlights that antisolvent injection is the decisive step that initiates either nanorod or nanoplatelet formation. To

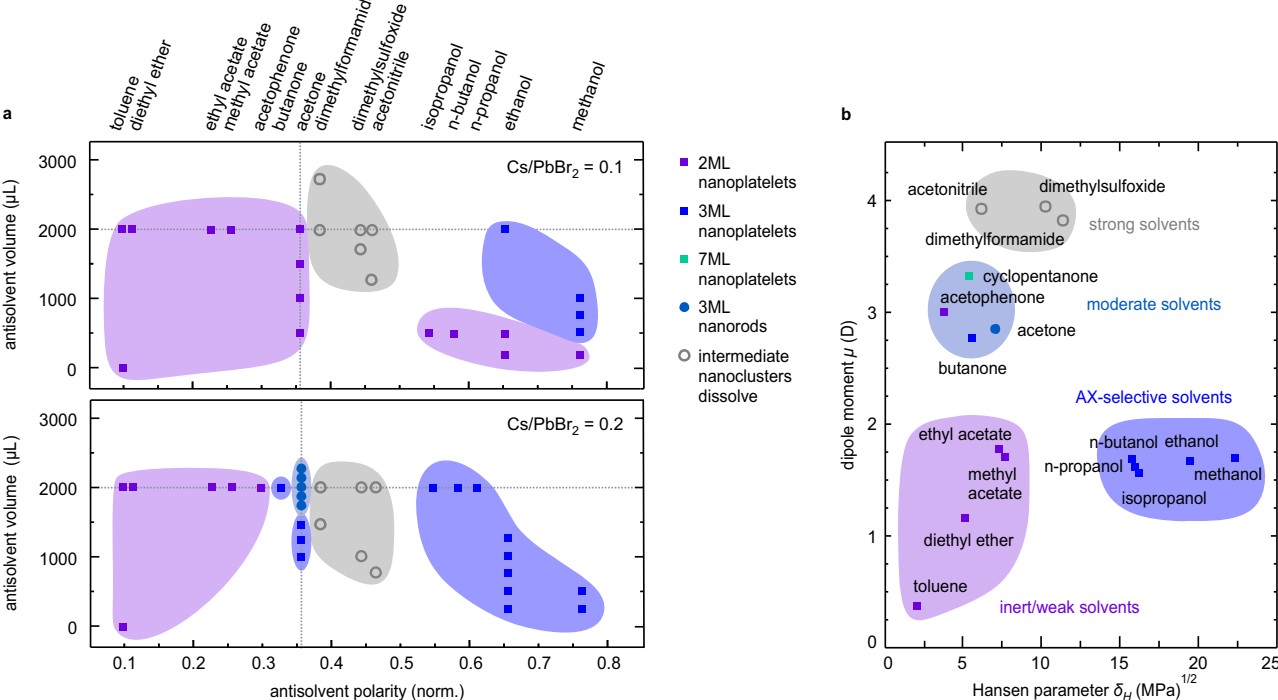

**Fig. 5 | Variation of antisolvent properties controls the dimensionality and thickness of anisotropic CsPbBr₃ nanocrystals. a** Shape and thickness of the obtained perovskite nanocrystals depending on Cs-oleate to PbBr₂ precursor ratio, relative antisolvent polarity and antisolvent volume. Acetone, the antisolvent used in the standard procedure of the nanorod and nanoplatelet synthesis and the typically injected volume are marked for reference (grey dashed lines). The type of nanocrystal product from each synthesis is indicated in the legend. **b** Shape anisotropy and ML (monolayer) thickness of perovskite nanocrystals classified by antisolvent properties (inert, weak, moderate, strong, and AX-selective solvents), legend as given in (**a**). Inert and weak solvents yield 2ML nanoplatelets, AX-selective solvents yield 3ML nanoplatelets. Strong solvents dissolve intermediate nanoclusters and do not yield any perovskite. Moderate solvents yield a variety of reaction products, with only acetone producing 3ML nanorods. Source data are provided as a Source Data file.

understand the role of the antisolvent in more detail, we screened other potential antisolvents, varying the antisolvent polarity and volume fraction (Fig. 5, Supplementary Note 6, Supplementary Tables 11 and 12, and Supplementary Fig. 28). Solvents with polarity equal to or less than acetone also result in 2ML nanoplatelets, provided the volume fraction is similar (Fig. 5a). At very high polarities, i.e., for alcohols, the platelet shape remains, but the thickness increases to 3ML nanoplatelets. Furthermore, an intermediate polarity regime exists, where the intermediates dissolve without forming a stable perovskite phase. Nanorods were only obtained under very specific conditions by using acetone as an antisolvent (Fig. 5a). We, therefore, further elucidated the special role of acetone in comparison to other antisolvents. For this purpose, we adopt the idea of classifying solvents according to their dipole moment, Hansen hydrogen bonding parameter, and donor number[40]. Solvents can be classified as inert, weak, moderate, strong, or AX-selective according to their effect on the ion solubility of lead halide perovskites. We find a clustering of synthesis products (Fig. 5b) in this parameter space, which coincides with the solvent classes. Weak solvents (diethyl ether, ethyl acetate, methyl acetate) and small amounts of acetone accelerate 2ML nanoplatelet formation, reducing reaction times (Supplementary Fig. 25) while monodispersity and optical properties of the nanoplatelets are maintained. Strong solvents, mainly characterised by a large dipole moment and/or high donor number (dimethylformamide, dimethylsulfoxide and acetonitrile), immediately dissolve the intermediate nanoclusters and prevent further LHP growth. Owing to a high electron-pair donor ability, these Lewis basic solvents strongly coordinate Pb²⁺ ions, inhibiting perovskite crystallisation[37]. Solvents with a high Hansen hydrogen bonding parameter, such as alcohols, are termed AX-selective due to their ability to coordinate organic A cations[40]. In our case, they also coordinate the weakly bound oleylamine ligands[48] from the surface of

the intermediate nanoclusters. Thereby, they enable the formation of thicker (3ML) nanoplatelets with very narrow, blue emission (462 nm, FWHM = 14 nm, Supplementary Fig. 29), likely via seeded growth from intermediate nanoclusters[56]. By reducing the injected volume of AX-selective solvents, again, 2ML nanoplatelets are obtained, suggesting that the hydrogen bonding ability, as well as the polarity of the solvent mixture, allows for ML thickness control. This is remarkable, as previous experiments mainly focused on the precursor ratio, specifically the A/B cation ratio, to control the resulting nanoplatelet thickness[15,38,41].

Moderate solvents like acetone occupy a narrow region in the $\mu$-$\delta_H$ diagram (Fig. 5b), characterised by an intermediate hydrogen bonding ability and dipole moment. This combination makes acetone rather unique. On the one hand, acetone destabilizes the intermediate nanoclusters moderately. On the other hand, acetone is well known to allow for an efficient transfer of ligand-coated nanocrystals from polar to nonpolar solvents by mediating surface tension mismatch in colloidal dispersions[57]. Both properties are needed for the nanorod synthesis, which proceeds via rapid mesophase formation according to our SAXS data. Indeed, other moderate antisolvents (butanone, cyclopentanone, acetophenone) which were added to the screening after analysis of the solubility parameter space, led to the formation of a variety of different perovskite nanocrystals, including 7ML nanoplatelets and 5ML nanorods, see Supplementary Fig. 30.

The broad screening of synthesis conditions combined with precise structural information during synthesis allowed us to obtain thickness and shape control of the final nanocrystals with excellent colour-tunability (Supplementary Note 7 and Supplementary Fig. 31). Furthermore, we achieved control of nanorod aspect ratio and lateral nanoplatelet size, shown in Supplementary Figs. 32 and 33. The as-synthesised nanocrystals showed PLQY values up to 42%

(Supplementary Fig. 34) and good stability, maintaining the initial PLQY for over three weeks, as shown in Supplementary Fig. 35. We expect that thicker nanorods and nanoplatelets could be synthesised via an analogous approach by adapting the precursor ratio (Supplementary Fig. 36) and the types of ligands to form larger intermediate nanoclusters. Employing similar moderate solvents, mesophase formation could be controlled to favour nanorod growth.

In conclusion, combining in situ optical spectroscopy and X-ray scattering, we reveal the formation mechanism of anisotropic $CsPbBr_3$ nanocrystals. We find that crystalline intermediate nanoclusters form immediately upon precursor mixing and play a crucial role in the growth of the final LHP nanocrystals. The intermediate nanoclusters serve either as reservoirs for the growth of nanoplatelets or as building blocks in mesophase-templated nanorod growth, depending on the antisolvent. In the first case, nanoclusters gradually dissolve and contribute to the formation of quasi-2D nanoplatelets as the thermodynamically most stable shape of LHP nanocrystals at low Cs-oleate to $PbBr_2$ precursor ratio and moderate antisolvent volume. In the second case, the antisolvent has a more profound impact and induces the formation of a well-ordered mesophase, resulting in a quasi-1D nanorod shape of LHP nanocrystals that is predefined by the geometry of the mesophase. Formation of nanorods is only observed when ketones such as acetone are used as antisolvent. We identify the dipole moment and Hansen hydrogen bonding parameter of the antisolvent as the most critical parameters in determining the final LHP nanocrystal shape. Solvents with moderate dipole moment and hydrogen bonding add further control in addition to established ligands and precursor ratios, providing room for extending the range of shape and size control of LHP nanocrystals.

## Methods

### Nanocrystal synthesis
First, the Cs-oleate precursor was prepared by dissolving $Cs_2CO_3$ (0.1 mmol, 32.6 mg) in oleic acid (10 mL) while stirring at 85 °C for up to three hours until a clear solution was obtained. Similarly, the $PbBr_2$ precursor was prepared by dissolving $PbBr_2$ (0.1 mmol, 36.7 mg) in toluene (10 mL), oleic acid, and oleylamine (100 $\mu$L each) under stirring at 85 °C for up to three hours until a clear solution was obtained. Precursors were stored under ambient conditions. Syntheses were carried out in ambient air at 20−40% humidity and at room temperature (25 °C). For the automated synthesis, a certain volume of $PbBr_2$ precursor (300 $\mu$L for 3ML nanorods, 400 $\mu$L for 2ML nanoplatelets) was added to the glass capillary of the custom-built in situ reaction cell. Under vigorous stirring, the Cs-oleate precursor (30 $\mu$L for 3ML nanorods, 20 $\mu$L for 2ML nanoplatelets) was injected with a motorised syringe pump. After 10 s, the antisolvent acetone (400 $\mu$L for 3ML nanorods, 267 $\mu$L for 2ML nanoplatelets) was injected through a second motorised syringe pump. The reaction mixture was stirred for up to 8 min and then centrifuged for 3 minutes (Eppendorf MiniSpin Plus, 1075 $\times g$), serving as the final purification. The supernatant was discarded, and the precipitate, containing the anisotropic $CsPbBr_3$ nanocrystals, was redispersed in n-hexane (500 $\mu$L) for further ex-situ characterisation. Precursor and antisolvent volumes for manually conducted ex-situ syntheses are listed in Supplementary Table 12. Chemicals are listed in Supplementary Table 13.

### WAXS, SAXS and PXRD
In situ WAXS data were recorded at the second experimental hutch EH2 of beamline P07[58] (PETRAIII, DESY, Hamburg) at an X-ray energy of 103.8 keV (wavelength $\lambda = 0.1195$ Å). The beamline setup[59] uses a helium-filled sample chamber for a reduced air scattering background. A custom-built reaction cell with magnetic stirring and reagent injection served as the sample environment, as described in detail in Supplementary Fig. 7. A XRD 4343CT detector (Varex Imaging) at 0.77 m distance was used with 0.5 s exposure time per frame. In situ SAXS data

were recorded at beamline P62[60] (PETRAIII, DESY, Hamburg) at an X-ray energy of 20 keV (wavelength $\lambda = 0.62$ Å). An Eiger2 X 9M detector (Dectris) in vacuum was used at 1.873 m distance with 50 ms exposure time per frame. The detector positions were calibrated using $LaB_6$ (WAXS) or silver behenate (SAXS) using pyFAI[61]. WAXS and SAXS data were transformed to intensity as a function of scattering vector $q = \frac{4\pi}{\lambda}\sin\theta$. $2\theta$ is the scattering angle. The signal of toluene and the antisolvent were subtracted as a background, scaled by the respective ratios.

WAXS data were fitted using a Pseudo-Voigt peak on a linear background between 1.92 and 2.28 Å$^{-1}$. The peak position, height, FWHM, Lorentz/Gauss ratio and two background parameters were allowed to vary. A instrumental resolution of $(3.342 \pm 0.008) \cdot 10^{-2}$ Å$^{-1}$ was estimated by fitting a reflection of $LaB_6$ at 2.11 Å$^{-1}$ and subtracted from the fitted FWHM as $\Delta q = \sqrt{FWHM_{fitted}^2 - FWHM_{LaB6}^2}$. The crystallite size $d_{domain}$ was calculated as $d_{domain} = \frac{2\pi}{\Delta q}$. The peak intensity was extracted by summing the intensities between 2.085 and 2.105 Å$^{-1}$. The reduced pair distribution function (PDF) $G(r)$ was obtained from WAXS data between 1.8 Å$^{-1}$ and 14.3 Å$^{-1}$ using PDFgetX3[62] as described in Supplementary Note 2.

SAXS data were fitted using SasView[63] with a two-component model, describing the $PbBr_2$ precursor as spherical core-shell micelles with a hard sphere structure factor, and the growing intermediate nanoclusters as ellipsoids. Ellipsoid diameter $d$ and length $l$, scale factors for precursor and nanoclusters, scale$_{PbBr2}$ and scale$_{NC}$, and a constant background term were allowed to vary. The criteria for model selection and alternative models are described in Supplementary Note 3. The number density of nanoclusters was calculated as $\rho = \frac{scale_{NC}}{\langle V \rangle}$, where $\langle V \rangle$ is the mean nanocluster volume.

Laboratory SAXS and PXRD were recorded at the chair for Soft Condensed Matter at LMU Munich using two setups with molybdenum $K_\alpha$ microfocus sources ($\lambda = 0.71$ Å)[64]. SAXS was recorded using the custom-built reaction cell. The detector for SAXS was a Pilatus3 R 300K (Dectris) at 1 m distance or a Pilatus 100K at 0.78 m distance. PXRD was recorded in transmission mode with samples dropcasted on adhesive tape. Here, a Pilatus 100K detector at 0.23 m distance was raster-scanned perpendicular to the X-ray beam.

Nanocrystal structures were drawn using VESTA[65].

### In situ PL spectroscopy
In situ PL spectra were recorded in the same custom-built reaction cell, simultaneously with SAXS and WAXS measurements. A 385 nm LED (Roschwege RSW-P01-385-2) filtered by a 10 nm FWHM band-pass filter (Thorlabs FBH380-10) was coupled into a reflection probe fibre (Thorlabs RP20) as the excitation. The emitted light captured by the reflection probe fibre was passed through a 400 nm long-pass filter (Thorlabs FELH0400) and recorded with a CCD-based UV-visible spectrometer (Ocean Optics Flame-S for in situ SAXS/PL, Ocean Insight QE-Pro for in situ SAXS/WAXS/PL measurements). The peak position, height, and FWHM were tracked and the results from subsequent frames were binned for further analysis. Emission at wavelengths smaller and larger than 450 nm was analysed by summing the intensities in bins from 415−450 nm and 450−470 nm, respectively.

Additional in situ PL spectra of 2ML nanoplatelet syntheses were recorded in a qpod 3 temperature-controlled cuvette holder (Quantum Northwest) with 1 s temporal resolution. A 365 nm LED (Thorlabs M365FP1) was coupled into a reflection probe fibre (Thorlabs RP20) as the excitation. The emitted light captured by the reflection probe fibre was passed through a 390 nm long-pass filter (Chroma Technology ET390LP) and recorded with a CCD-based UV-visible spectrometer (Thorlabs CCS200/M). For further analysis, PL intensities of background-corrected spectra were integrated in the range between 400−500 nm and normalised with respect to the maximum value.

**Ex situ absorbance, PL, PLE spectroscopy and PLQY measurements**

Ex situ PL spectra, PLE spectra and UV-Vis absorbance spectra of purified and diluted $CsPbBr_3$ nanocrystals were measured in a commercial FluoroMax-4Plus spectrometer equipped with a xenon arc lamp and an F-3031 transmission accessory (HORIBA Scientific). The excitation wavelength for PL spectra was set to 380 nm. Absolute PLQY values were determined with a Quanta-$\phi$ F-3029 integrating sphere (HORIBA Scientific).

**Electron Microscopy**

Annular dark field scanning transmission electron microscopy (ADF-STEM) images were recorded using a probe-corrected FEI Titan Themis 60-300 operated at an acceleration voltage of 300 kV with a semi-convergence angle of 16.6 mrad. The inner and outer collection angles of the annular dark field detector were 33 and 198 mrad. No further filtering was applied to the obtained micrographs. Specimen preparation was carried out by drop casting onto TEM grids (Quantifoil R2/2, 2 nm ultrathin carbon). TEM images at lower magnifications were recorded on a JEOL JEM-1100 microscope operated at an acceleration voltage of 80 kV. Specimen preparation was carried out by drop casting onto TEM grids (Electron Microscopy Sciences, Cu with 10/1 nm Formvar/carbon).

## Data availability

The raw PL, SAXS, WAXS, and TEM data analysed in this study are available in the Open Data LMU - Physics repository https://doi.org/10.57970/nb26d-cak63[66]. Source data are provided with this paper.

## Code availability

The code for in situ PL, SAXS, and WAXS data analysis is available in the Open Data LMU - Physics repository https://doi.org/10.57970/nb26d-cak63[66].

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

## Acknowledgements

This work was supported by the Bundesministerium für Bildung und Forschung (BMBF) via the projects 05K19WMA and 05K22WMA (B.N.), by the European Research Council Horizon 2020 through the ERC Grant Agreement PINNACLE (759744) (A.S.U.), by the Deutsche Forschungsgemeinschaft (DFG) under Germany's Excellence Strategy EXC 2089/1-390776260 (Excellence Cluster 'e-conversion') (A.S.U., K.M.-C.) and the Bavarian State Ministry of Science, Research and Arts through the grant 'Solar Technologies go Hybrid (SolTech)' (B.N., A.S.U.). Experiments were carried out in part at PETRAIII at the Deutsches Elektronen-Synchrotron DESY (Hamburg, Germany), a member of the Helmholtz Association (HGF) at the P62 and P07 beamlines (proposal numbers I-20200518, I-20220308 and II-20200002), and were supported through the Maxwell computational resources. We thank Quinten Akkerman for helpful comments.

## Author contributions

K.F. and N.A.H. contributed equally to this work. A.S.U. and B.N. conceived and supervised the project. N.A.H., C.L., and V.M. synthesised the nanocrystals. K.F., C.L., X.S., S.H., A-C.D., O.G., and B.N. designed the simultaneous SAXS/WAXS/PL experiment, and K.F., N.A.H., and C.L. carried out the measurements. N.A.H. recorded TEM images, absorption, PL, and PLE spectra and analysed the respective data. T.L. and B.M. recorded ADF-STEM images under the supervision of K.M.-C.. T.L. analysed the ADF-STEM images. K.F. and N.A.H. carried out laboratory SAXS measurements. K.F. analysed the SAXS, WAXS, PDF, and in situ PL data. K.F., N.A.H., A.S.U., and B.N. wrote the manuscript. All authors discussed the results and commented on the manuscript.

## Funding

## Competing interests

The authors declare no competing interests.
