## [Peer Review file · Nature Communications]

Antisolvent controls the shape and size of anisotropic lead halide perovskite nanocrystals

Corresponding Author: Dr Bert Nickel

Version 0:

Reviewer comments:

Reviewer #1

(Remarks to the Author)

This manuscript projected the captured growth process of platelet and rod-shaped nanostructures of CsPbBr₃. The main emphasis here to see the changed via SAXS data. No doubt the information is important for the community. However, several issues are speculative and mostly presented in main manuscript via scheme rather than with data. Hence, this reviewer is confused and unable to recommend this article for NC.

Comments:

- (1) The title says shape and size of anisotropic, and there is not much on either size or shape evolutions and all products are reported.
- (2) The shape derivation details should be supported with microscopic images, but only two set of shapes are shown. Rods are also not clear at this stage.
- (3) Entire story should be specialized to SAXS study rather the generalized titles.

With these points, I do not see much to look in this manuscript which is too much specialized.

Reviewer #2

(Remarks to the Author)

This manuscript deals with the formation mechanisms of anisotropic lead halide perovskite nanocrystals, 1D nanorods and 2D nanoplatelets, by combining in situ X-ray scattering and photoluminescence spectroscopy. The authors found that the divergent anisotropy is induced by ensuing antisolvent addition: The intermediate nanoclusters are driven into a dense hexagonal mesophase, where they fuse to form nanorods. Contrastingly, nanoplatelets grow freely dispersed from dissolving nanoclusters, stacking subsequently in lamellar superstructures. Shape and size control of the nanocrystals are determined primarily by the antisolvent's dipole moment and Hansen hydrogen bonding parameter. Since this work exploited the interplay of antisolvent and organic ligands enable more complex nanocrystal geometries, I recommend it for publication after revisions as listed below:

1. The stability of the as-synthesized 1D nanorods and 2D nanoplatelets need to be analyzed.
2. The PLQY of the as-synthesized 1D nanorods and 2D nanoplatelets need to be analyzed.
3. It was read that "A reaction glass was loaded with PbBr₂-precursor solution and Cs-oleate was immediately added under vigorous stirring. After 10 s, the antisolvent acetone..." The readers may wonder is there any nanocrystals formed and PL emission seeing before the adding of the antisolvent acetone.
4. Could the authors comment on the changes in FWHM of the the as-synthesized 1D nanorods and 2D nanoplatelets?
5. Fig. 1 | Synthesis and characteristics of the anisotropic perovskite nanocrystals. It was noted that there is a purification process after 60-120s. Please add some details on this purification process, which is important for the sample characterizations.

Reviewer #3

(Remarks to the Author)
see attached pdf

Reviewer #4

(Remarks to the Author)

The manuscript by Frank and co-workers entitled "Antisolvent controls the shape and size of anisotropic lead halide perovskite nanocrystals", reports on the formation mechanisms of nanorods (NRs) and nanoplatelets (NPLs) of lead halide perovskite colloidal nanocrystals (NCs) synthesized by the solvent/antisolvent method at room temperature. The synthesis is monitored by in situ small and wide-angle X-ray scattering (SAXS, WAXS) and photoluminescence (PL) spectroscopy and supported by transmission electron microscopy (TEM). The authors show that for both end products, at the first step of mixing the PbBr₂ and Cs-oleate precursors (in the presence of oleic acid and oleylamine ligands), an initial specie of nanoclusters is formed with PL centered around 450nm. Then, acetone, an antisolvent is introduced into the reaction mixture, inducing a dense mesophase, causing the fusion of the initial nanoclusters to form NRs. In contrast, NPLs are formed by the dissolution of the initial nanoclusters and grow freely in the synthesis medium. Subsequently, stacking in lamellar superstructures. Most interestingly, the authors investigate a comprehensive list of different antisolvents, showing control over shape and size of the NCs which is determined primarily by the antisolvent's dipole moment and Hansen hydrogen bonding parameter. The authors give a broad parameter space of volumes different solvents and precursor's concentrations.

I have a few concerns I ask the authors to address:

1. Cs₄PbBr₆ phase, typically an unwanted non emissive by product with an absorption peak at 315nm was not monitored.
2. Water have been shown to promote the formation of NRs but is not discussed here (for instance, "Efficient Interfacial Synthesis Strategy for Perovskite CsPbBr₃ Nanorods in the Biphasic Solution", Adv. Mater. Technol. 2022, 7, 2200131, DOI: 10.1002/admt.202200131).
3. The authors don't address control over the aspect ratio of the NRs.
4. The authors may add a reference: "Shape-controlled synthesis of CsPbBr₃ nanorods with bright pure blue emission and high stability", J. Mater. Chem. C, 2024, 12, 4234-4242, DOI:10.1039/D3TC04681H.

Although X-ray scattering is not in the field of my expertise and I'm not able to comment on the quality of these results. The PL and TEM results are convincing enough and the overall experimental design and presentation of the results is very good and comprehensive. In light of all that, I can recommend the publication of this manuscript in Nature Communications after the above comments are addressed.

Version 1:

Reviewer comments:

Reviewer #1

(Remarks to the Author)

Authors addressed comments as per the best. I have no more to say.

Reviewer #2

(Remarks to the Author)

Since this manuscript is properly revised according to the comments, this reviewer recommend it to be published as it is.

Reviewer #3

(Remarks to the Author)

I am very satisfied by the author revisions in response to my referee report, as well as those provided by other reviewers. I believe the manuscript is ready for publication.

Reviewer #4

(Remarks to the Author)

The authors sufficiently answered all of my concerns and other comments raised by the other referees. Thus, I fully recommend this paper for publication in Nature Communications.

RESPONSE TO REVIEWERS' COMMENTS

Reviewer 1:

This manuscript projected the captured growth process of platelet and rod-shaped nanostructures of CsPbBr₃. The main emphasis here to see the changed via SAXS data. No doubt the information is important for the community. However, several issues are speculative and mostly presented in main manuscript via scheme rather than with data. Hence, this reviewer is confused and unable to recommend this article for NC. ... With these points, I do not see much to look in this manuscript which is too much specialized.

Response: Thank you for assessing these results as important. We have clarified how our claims are related to experimental data, and we added new data where needed. Please see below for details.

✓R1Q1: The title says shape and size of anisotropic, and there is not much on either size or shape evolutions and all products are reported.

Response: We agree that the title is a very delicate aspect of a paper. Our title, "Antisolvent controls the shape and size of anisotropic lead halide perovskite nanocrystals," is intended to signal to the reader that (in contrast to popular understanding) antisolvent plays a huge role in determining the final shape and size of halide perovskite nanocrystals. Indeed, in our synthesis, we identify conditions for anisotropic shapes such as nanorods, nanoplatelets, and slightly anisotropic nanoclusters. The size variation with different synthesis conditions is also presented and discussed. Since these materials show quantum confinement, the discussion focuses on thickness in the case of the nanoplatelets and diameter in the case of nanorods. We admit that lateral size is only discussed briefly, e.g., in the context of nanoplatelet stack formation. But for the sake of title compactness, we would like to stay here with "size" rather than "the smallest dimension of nanoparticles which determines quantum confinement effects".

Action: We have added a discussion on the overall obtainable shapes and sizes and provide additional data on tunable aspect ratio of 3ML CsPbBr₃ nanorods in Supplementary Fig. 32 and tunable lateral sizes of the 2ML CsPbBr₃ nanoplatelets in Supplementary Fig. 33. **Changes to the manuscript:** Added "Furthermore, we achieve control of nanorod aspect ratio and lateral nanoplatelet size over a wide range, shown in Supplementary Figs. 32 and 33." (page 9)

✓R1Q2: The shape derivation details should be supported with microscopic images, but only two set of shapes are shown. Rods are also not clear at this stage.

Response: We agree that electron microscopy can be helpful in understanding the shape derivations of nanostructures. We have included more electron microscopy images in the manuscript and Supplementary Information. In the main manuscript, we show electron microscopy of 3ML nanorods and 2ML nanoplatelets. Additionally, we show the electron microscopy images of prolate intermediate nanoclusters, which form in the first step of the synthesis (Fig. 3 and Supplementary Fig. 20). We also provide the nanocrystal structures obtained without antisolvent (Supplementary Fig. 26). Further electron microscopy images included in the Supplementary Information are those of 3ML nanoplatelets (Supplementary Fig. 29), 5ML nanorods and 7ML nanoplatelets (Supplementary Fig. 30), 3ML nanorods with tunable aspect ratio (Supplementary Fig. 32) and 2ML nanoplatelets with tunable edge length (Supplementary Fig. 33). We also added high-resolution images with indexing of lattice planes (Supplementary Fig. 2, see below R3Q4b).

Action: **Changes to the manuscript:** We have included high-resolution TEM images of rods and platelets (Supplementary Fig. 2, see R3Q4b).

✓R1Q3: Entire story should be specialized to SAXS study rather the generalized titles.

Response: We obtain our results by a multi-method approach, including electron microscopy and optical spectroscopy, and we aim for the broad perovskite community. We improved the two paragraphs which describe the rod and platelet formation for clarity. We agree that the SAXS part is especially strong and convincing. We are considering the advice to discuss certain aspects of the X-ray experiment in a subsequent technical paper for the X-ray community.

Reviewer 2:

This manuscript deals with the formation mechanisms of anisotropic lead halide perovskite nanocrystals, 1D nanorods and 2D nanoplatelets, by combining in situ X-ray scattering and photoluminescence spectroscopy. The authors found that the divergent anisotropy is induced by ensuing antisolvent addition: The intermediate nanoclusters are driven into a dense hexagonal mesophase, where they fuse to form nanorods. Contrastingly, nanoplatelets grow freely dispersed from dissolving nanoclusters, stacking subsequently in lamellar superstruc-

tures. Shape and size control of the nanocrystals are determined primarily by the antisolvent's dipole moment and Hansen hydrogen bonding parameter. Since this work exploited the interplay of antisolvent and organic ligands enable more complex nanocrystal geometries, I recommend it for publication after revisions as listed below:

Response: Thank you for recommending our work for publication. We believe that the revisions below will meet your expectations.

✓R2Q1: The stability of the as-synthesized 1D nanorods and 2D nanoplatelets need to be analyzed.

Response: We analysed the stability of 1D nanorods and 2D nanoplatelets in colloidal dispersion by tracking their absorbance and PL spectra over a period of three weeks. For each measurement, the integrated PL intensity was divided by the sample absorbance at the excitation wavelength ($\lambda = 400$ nm) and normalized with respect to the value measured immediately after synthesis, i.e., on day 0, to obtain the relative PLQY. We find that 2ML nanoplatelets, 3ML nanorods, and 3ML nanoplatelets demonstrate excellent colloidal stability throughout the observed timeframe. Most importantly, PL spectra of aged samples (day 23) are almost identical to those of fresh samples (day 0).

Action: Changes to the manuscript: We added a plot of the temporal evolution of relative PLQY values of 2ML and 3ML nanoplatelets and 3ML nanorods in Supplementary Fig. 35, including representative PL spectra of fresh and aged colloidal solutions for comparison and refer to the PLQY and stability in the text (page 9).

✓R2Q2: The PLQY of the as-synthesized 1D nanorods and 2D nanoplatelets need to be analyzed.

Response: We measured and analysed the absolute PLQY of different batches of as-synthesised 2ML nanoplatelets, 3ML nanorods, and 3ML nanoplatelets dispersed in hexane with an integrating sphere setup. On average, the PLQY is (8.4 ± 4.0) % for 2ML nanoplatelets, (9.0 ± 4.0) % for 3ML nanorods and (25.3 ± 10.1) % for 3ML nanoplatelets. Similar PLQY values in anisotropic CsPbBr₃ nanocrystals have been shown to result from the presence of surface lead and bromide vacancies. A post-synthetic treatment of nanocrystals with enhancement solution containing PbBr₂, oleylamine, and oleic acid in hexane is an effective means for repairing surface defects in as-synthesised perovskite nanocrystals. Here, such a treatment boosts PLQY up to 26% for 2ML NPLs, 58% for 3ML nanorods and 70% for 3ML nanoplatelets.

Action: Changes to the manuscript: We have included a plot showing the absolute PLQY values of as-synthesised and enhanced 3ML nanorods, 2ML and 3ML nanoplatelets in Supplementary Fig. 34.

✓R2Q3: It was read that "A reaction glass was loaded with PbBr₂-precursor solution and Cs-oleate was immediately added under vigorous stirring. After 10 s, the antisolvent acetone..." The readers may wonder is there any nanocrystals formed and PL emission seeing before the adding of the antisolvent acetone.

Response: Immediately upon addition of the Cs-precursor to the PbBr₂-precursor, nanoclusters form with dimensions of roughly $2 \times 2 \times 5$ nm², and with an emission wavelength of 446 nm. In the conclusion, we write accordingly (page 9): "*We find that crystalline intermediate nanoclusters form immediately upon precursor mixing and play a crucial role in the growth of the final LHP nanocrystals. The intermediate nanoclusters serve either as reservoirs for the growth of nanoplatelets or as building blocks in mesophase-templated nanorod growth, depending on the antisolvent.*".

R2Q4: Could the authors comment on the changes in FWHM of the as-synthesized 1D nanorods and 2D nanoplatelets?

✓Response: For as-synthesised quasi-2D nanoplatelets and quasi-1D nanorods, the optical properties including FWHM are dictated by quantum confinement effects stemming from the smallest nanocrystal dimension. Nanorods presented in this manuscript experience strong quantum confinement in two dimensions (height and width), and the size inhomogeneity of both dimensions is reflected in the PL profile (FWHM ≈ 18 nm). Nanoplatelets experience strong confinement in one dimension (thickness) since the lateral dimensions fall outside the (strong) confinement regime of CsPbBr₃. As the inhomogeneity of the lateral dimensions does not contribute significantly to inhomogeneous broadening and there is virtually no inhomogeneity in the thicknesses due to surface energy minimisation, the PL profiles of the nanoplatelets are significantly narrower (FWHM $\approx 11-14$ nm).

✓R2Q5: Fig. 1 — Synthesis and characteristics of the anisotropic perovskite nanocrystals. It was noted that there is a purification process after 60-120s. Please add some details on this purification process, which is important for the sample characterizations.

Response: The reaction mixture is purified once after the synthesis is completed. Purification is done by centrifugation of the reaction mixture (4000 rpm, $r = 60$ mm, 3 minutes). The supernatant is discarded, and the yellow precipitate containing nanorods or nanoplatelets is redispersed in n-hexane.

Action: Changes to the manuscript: We have modified the section "Synthesis scheme" and the Methods section "Nanocrystal synthesis" to clarify that the centrifugation of the reaction mixture and resuspension of the precipitate serves as the final purification step. We have removed the term "purification" from Fig. 1, since the different nanocrystal shapes are already observed prior to purification.

Reviewer 3:

This is a straightforward manuscript giving insight into the growth of anisotropic CsPbBr₃ nanorods and platelets using in situ X-ray and spectroscopy techniques. I think the manuscript can be improved by providing more clarity on several points, discussed below, and I also think more citations and a stronger connection to comparable research can be provided. Once these issues are addressed, I think the manuscript would be ready for publication.

Response: We would like to thank the Reviewer for providing interesting references and identifying critical points. We have extended our analysis and are now more precise in the wording. Please see below for details.

✓R3Q1: Page 3, Line 63: There are studies that demonstrate slow reaction kinetics, and maintain the cuboid shape of a conventional fast synthesis. I encourage the authors to read and cite Chem. Mater. 2019, 31, 20, 8551–8557

Response: We thank the reviewer for making us aware of this publication.

Action: Changes to the manuscript: We have included the suggested reference²⁹ in the introduction.

✓R3Q2: Page 3, lines 80 and 83: What conditions? The wording here is a little confusing as it is left a little vague.

Response: We are sorry if the wording here in the introduction was vague. It is explained in more detail later in the main manuscript and in the conclusion. To clarify this early on in the manuscript, we have changed the text in several parts:

Action: Changes to the manuscript:

- Amended introduction: *"Subsequent injection of large volumes of the antisolvent acetone at higher Cs to PbBr₂ precursor ratio induces the formation of a dense, hexagonal mesophase of the intermediates, wherein they fuse to form nanorods."*
- Amended conclusion: *"In the second case, the antisolvent has a more profound impact and induces the formation of a well-ordered mesophase, resulting in a quasi-1D nanorod shape of LHP nanocrystals that is predefined by the geometry of the mesophase. Formation of nanorods was only observed when ketones such as acetone were used as antisolvent."*

✓R3Q3: Page four, figure 1. Nanorods are shown to have dimensions of 1.8 nm (width) and 15 nm lengths. The emission peak for these CsPbBr₃ nanorods is 460 nm. The emission seems much more red-shifted than I believe has been reported in other studies of quantum-confined nanorods. It would be helpful to compare these samples with the other examples in the literature, especially e.g. Nano Lett. 2022, 22, 20, 8355–8362 and ACS Nano 2022, 16, 5, 8318–8328 – I wonder if there is a systematic difference in how the authors of these different manuscripts are determining rod width.

Response: Relating physical size and optical shift is challenging since the smallest size, which dominates the confinement, needs to be determined with high precision. Usually, SEM does not provide enough resolution, and TEM yields the 2D projection of a 3D object, i.e., the height is not easily resolvable. In order to obtain the most accurate estimate for the size and dimensionality of our nanocrystals, we combine several methods. For the nanorods, we determine width by manual measurement of nanorod thickness in TEM and ADF-STEM images (Supplementary Fig. 1 a), by the repeat distance in TEM images obtained from the FFT (Supplementary Fig. 1 b), and by fitting SAXS data (diameter between 1.7 and 2.3 nm, Supplementary Tables 1 and 6). The values deviate by less than 0.6 nm, which justifies the term "3ML" for the nanorod diameter, in our opinion. Therefore, the 460 nm emission observed here can be correlated to a dimension of approximately 3-4 monolayers.

In the studies mentioned by the Reviewer, the authors observed nanostructures with a short dimension of the order of 3.4–3.9 nm, i.e., 6 monolayers, see e.g. Figure 2 in Nano Lett. 2022, 22, 20, 8355–8362. We think that it might be quite possible that these nanocrystals are thinner than they appear in TEM, due to a hypothetical "belt-like" structure, similar to previous studies⁴². Thus, the discrepancy noted by the referee is quite likely much smaller than noted (if present at all).

Action: Changes to the manuscript: In order to explain better the size determination by TEM, we added a TEM image and respective FFT of 3ML nanorod assemblies to Supplementary Fig. 1 and included the following caption:

"Size distribution of CsPbBr₃ 3ML nanorods and 2ML nanoplatelets. a, 3ML nanorods have an average diameter of (1.9 ± 0.2) nm and a length of (15.0 ± 2.2) nm, also shown in Fig. 1 c. b TEM image of assembled NRs and the respective FFT image, confirming a regular spacing (4.83 nm) of 3ML nanorods. Nanorods lay flat on the TEM substrate. c, 2ML nanoplatelets have an average thickness of (1.3 ± 0.1) nm and square lateral dimensions of (15.6 ± 2.3) nm, as shown in Fig. 1 c. d, TEM image of stacks of edge-up oriented 2ML NPLs and the respective FFT image, confirming a regular stacking distance (4.16 nm) of 2ML nanoplatelets."

We refer to the SAXS fits in the text: *"Since TEM imaging can be ambiguous in distinguishing rod and platelet shape depending on the observed orientations^{41,42}, we additionally confirm the nanocrystal shape via small-angle X-ray scattering (SAXS) in solution (Supplementary Fig. 6)."* and added laboratory SAXS to the methods section.

✓R3Q4: Relatedly, the authors perform SAXS and WAXS, but do not show powder XRD patterns. Are the samples produced pure perovskite? Commonly, lead-deficient Cs₄PbBr₆ is produced during a synthesis, and may be expected to vary as a side-product across the parameter space being studied.

Response: The samples are indeed pure perovskite for the reaction conditions presented here. In situ WAXS measurements (Supplementary Fig. 10) and additional PXRD measurements performed in response to the reviewer request (Supplementary Fig. 4) show no reflections of lead-deficient Cs₄PbBr₆. Also, no CsBr-intermediate (which was identified in Chem. Mater. 2019, 31, 20, 8551–8557) was observed. Complementary absorption and photoluminescence excitation (PLE) spectra (Supplementary Fig. 5) of produced samples also do not exhibit a contrasting signal intensity at 315 nm, which would be expected if lead-deficient, non-emissive Cs₄PbBr₆ were produced as a side-product.

Action: Changes to the manuscript: We added Supplementary Fig. 4 (PXRD analysis), Supplementary Fig. 5 (absorption and PLE analysis), Supplementary Fig. 10 (in situ WAXS analysis). We refer to these figures in the main text: *"Powder X-ray diffraction (PXRD), (Supplementary Fig. 4) and photoluminescence excitation spectroscopy (PLE, Supplementary Fig. 5), show no non-emissive Cs₄PbBr₆ byproduct."* We added PXRD and PLE spectroscopy to the methods section.

✓R3Q4b: Can the authors correlate the XRD pattern of the product with the lattice planes that are resolved in the TEM images of nanorods and nanoplatelets?

Response: We analysed the high-resolution ADF-STEM images in more detail. In Supplementary Fig. 2, we identify and index lattice planes such as ((10 $\bar{1}$), (040), (24 $\bar{2}$)). The observed directions are consistent with an orthorhombic system in agreement with the XRD patterns.

Action: Changes to the manuscript: We added Supplementary Fig. 2 with ADF-STEM images of nanorods and nanoplatelets.

✓R3Q5: Additionally, can the authors identify which facet these rods "grown" along? Are all the nanorods oriented in the direction of a specific facet? This would require looking at the lattice planes on high resolution TEM images.

Response: We reinspected the high-resolution images and obtained better micrographs. The nanoplatelet and nanorod long axes correspond to the $\langle 100 \rangle$ axes in the cubic system (Supplementary Fig. 2). Often, several neighbouring nanorods lie on the same facet, i.e., columns of atoms of, e.g., $\langle 100 \rangle$ direction are visible (in cubic notation).

Action: Changes to the manuscript: Please see changes in response to R3Q4b.

✓R3Q6: Page 4 line 94 "The synthesis is conducted at room temperature in ambient conditions", can the

authors be more specific about “ambient conditions”? If it is referring to temperature, then it was already specified. In the case where ambient conditions in reference to the ambient atmosphere (i.e. oxygen, moisture etc.), there seems to be a discrepancy between the custom-built reaction cell being filled with Helium and it being under “ambient conditions”, could the authors please be more specific?

Response: We clarified that “ambient conditions” corresponds to room temperature (25 °C) and ambient air at a humidity of 20-40%. Only for the in situ WAXS experiments the in situ cell was placed in a helium atmosphere to reduce air scattering.

Action:

- Amended section “Synthesis scheme”: *“The synthesis is conducted in ambient conditions and can be described as a ligand-assisted spontaneous crystallisation.”*
- Amended methods: *“Precursors were stored under ambient conditions. Syntheses were carried out in ambient air at 20-40% humidity and at room temperature (25 °C).”*

✓R3Q7: Page 4, Line 100: It is not clear how the authors can deduce the role of Br versus Pb for determining product - these ions are always present in the same stoichiometric ratio by the choice of precursor (PbBr₂). Other work has identified that Pb plays little role in the synthesis (see Chem. Mater. 2019, 31, 20, 8551–8557). Perhaps the authors have done control studies that identify the Cs/Pb ratio, uniquely compared to the Cs/Br ratio?

Response: This is indeed an interesting question, and to answer it we have carried out additional syntheses. Using precursor solutions containing Pb (Pb-oleate) or Br (ZnBr₂) to vary the stoichiometric ratio of Pb/Br in the synthesis, we have studied the influence of Cs/Pb and Cs/Br ratio on the shape and size of resulting nanocrystals. We find that both the Cs/Pb and Cs/Br (and also Pb/Br) ratios impact the size and, therefore, the emission wavelength of the perovskite product, as shown in Supplementary Fig. 36. However, shape control, i.e., the formation of quasi-1D nanorods as opposed to the formation of quasi-2D nanoplatelets, is still only achievable by controlling the stability of intermediate perovskite nanoclusters through antisolvent engineering. As these results are quite interesting, we are now devoting resources to exploring this question further and hope to publish the results in a separate manuscript.

Action: We added Supplementary Fig. 36 to show the influence of Cs/Pb, Cs/Br and Pb/Br ratio on the shape and emission wavelength of produced perovskite nanocrystals. Furthermore, we replace “ratio of Cs⁺/Pb₂⁺ ions” with “ratio of Cs-oleate to PbBr₂ precursor” to acknowledge the interplay of different ion ratios for determining the anisotropic nanocrystal product.

✓R3Q8: Page 5, Line 140: How many units of PbBr₂ does the “precursor micelle” correspond to? Is it a crystalline cluster? This is a very interesting result. Is this reported in other works, and the authors are corroborating that finding? It would be very good to say one way or the other, and cite appropriately if so. What is the stoichiometry of that micelle?

Response: We determine the volume of the micellar core from SAXS analysis (Supplementary Table 5) to $(0.9 \pm 0.2) \text{ nm}^3$. According to the bulk density of PbBr₂ and its molecular weight, a precursor micelle contains approx. 9.6 ± 2.2 Pb atoms and in turn 19.2 ± 4.4 Br atoms. The distances between Pb and Br ions in these micelles were reinspected by a PDF analysis shown in Supplementary Fig. 15. The atomic distances are in agreement with an octahedral coordination. The core of these micelles is so small (only a few atoms in diameter) that the question of whether this is crystalline (or short-range ordered) is difficult to answer. In the optical range, the PbBr₂-precursor solution exhibits a strong absorption feature at 320–330 nm (Supplementary Fig. 14). This is similar to crystalline oleate/oleylamine-capped PbBr₂ nanocrystals or higher-coordinating multi-nuclear lead bromide species, suggesting that the stoichiometry of the micelle is conserved to PbBr₂. Since Pb²⁺ and Br⁻ ions are insoluble in toluene and require ligands for stabilization, such micelles, or micelle-like species, are a common feature of nanocrystal syntheses and have been discussed in various papers reporting on the syntheses of perovskite materials (^{16,19,20}).

Action: **Changes to the manuscript:** We adapted Fig. 3 c, which shows the PbBr₂ micelles. We added a comparison of WAXS data to crystalline PbBr₂ and a PDF analysis of the PbBr₂ precursor (Supplementary Fig. 15) as well as absorption spectra of the PbBr₂ precursor solution (Supplementary Fig. 14) and reference²⁰. We refer to the supporting figures in the text (page 5): *“A fit to the signal of the PbBr₂-precursor before mixing (solid line at $t = -0.5\text{s}$) shows that the PbBr₂-precursor is dispersed in the form of micellar nanoclusters, with 1.2 nm diameter PbBr₂-cores enclosed in ligand micelles of 2.8 nm diameter (see Supplementary Fig. 13*

and Supplementary Table 4). Each micelle contains on average 9.6 ± 2.2 PbBr_2 units (Supplementary Table 5). Absorption spectra of the PbBr_2 -precursor, shown in Supplementary Fig. 14, further corroborate the micellar structure. The internal structure of the PbBr_2 -core was further analyzed via the reduced pair distribution function (PDF) obtained from X-ray total scattering (Supplementary Fig. 15 and Supplementary Note 2). The distances between Pb and Br ions in these micelles are in agreement with an octahedral coordination.” We added PDF to the methods section.

✓R3Q8b: It seems quite important to understand if the precursor micelle is the template for the mesoscale hexagonal lattice that forms later. I think more clarity/discussion about this species, and the exact transformation into the next proposed stage of the growth would be very helpful for the manuscript. Please be clear about what is proven versus what is hypothesized but unsubstantiated.

Response: Nanoclusters form already after precursor mixing (i.e., prior to antisolvent injection) at the expense of micelles (Fig. 2 and Supplementary Fig. 19). The core diameter of the nanoclusters is 1.8–2.1 nm. Assuming stoichiometric LHP, this requires about 40 Pb ions and 120 Br ions (Supplementary Table 5), i.e., four to six PbBr_2 micelles per nanocluster. The template for the hexagonal mesophase is the intermediate nanoclusters, which is also confirmed by the centre-to-centre distance of the nanoclusters in the mesophase. The two paragraphs which discuss the transformation of intermediate nanoclusters to nanorods and nanoplatelets were improved for clarity. Now we write directly what is proven from each SAXS observation rather than first grouping several observations.

Action: Changes to the manuscript: We clarified the text (pages 6-7, please see also response to R3Q8), added Supplementary Fig. 19, and added “A comparison of the core sizes of the PbBr_2 precursor micelles and the nanoclusters reveals that initially 4-6 PbBr_2 micelles contribute to nucleation, and further material is incorporated during growth (Supplementary Table 5). As a consequence, we observe a decline of the number density of precursor micelles by 25% during nanocluster nucleation, shown in Supplementary Fig. 19.” (page 5)

✓R3Q9: Page 4, panel C: Why is the I001 signal not monotonic?

Response: The (001) reflection of the mesophase is at very low q and has several contributions. In the plot, we tried to separate the huge background signal, which underlies the (001) intensity, but this is intrinsically difficult since the background level drops in intensity with time. The non-monotonic behaviour here corresponds to less than 10% of total intensity change.

Action: Changes to the manuscript: We moved I_{001} to the supporting information. In the SI, we adapted Supplementary Fig. 22 to highlight the high background level around the (001) reflection, the change in position of the reflections, as well as the uncertainties of the values with error bars.

Reviewer 4:

The manuscript by Frank and co-workers entitled “Antisolvent controls the shape and size of anisotropic lead halide perovskite nanocrystals”, reports on the formation mechanisms of nanorods (NRs) and nanoplatelets (NPLs) of lead halide perovskite colloidal nanocrystals (NCs) synthesized by the solvent/antisolvent method at room temperature. The synthesis is monitored by in situ small and wide-angle X-ray scattering (SAXS, WAXS) and photoluminescence (PL) spectroscopy and supported by transmission electron microscopy (TEM). The authors show that for both end products, at the first step of mixing the PbBr_2 and Cs-oleate precursors (in the presence of oleic acid and oleylamine ligands), an initial specie of nanoclusters is formed with PL centered around 450nm. Then, acetone, an antisolvent is introduced into the reaction mixture, inducing a dense mesophase, causing the fusion of the initial nanoclusters to form NRs. In contrast, NPLs are formed by the dissolution of the initial nanoclusters and grow freely in the synthesis medium. Subsequently, stacking in lamellar superstructures. Most interestingly, the authors investigate a comprehensive list of different antisolvents, showing control over shape and size of the NCs which is determined primarily by the antisolvent’s dipole moment and Hansen hydrogen bonding parameter. The authors give a broad parameter space of volumes different solvents and precursor’s concentrations. I have a few concerns I ask the authors to address. ... Although X-ray scattering is not in the field of my expertise and I’m not able to comment on the quality of these results. The PL and TEM results are convincing enough and the overall experimental design and presentation of the results is very good and comprehensive. In light of all that, I can recommend the publication of this manuscript in Nature Communications after the above comments are addressed.

Response: Thank you for agreeing that our work is interesting and recommending it for publication. We have addressed your concerns in detail below.

✓R4Q1: Cs₄PbBr₆ phase, typically an unwanted non emissive by product with an absorption peak at 315nm was not monitored.

Response: We carried out further WAXS, PXRD, absorption and PLE analysis, as detailed in the response to R3Q4, and found no Cs₄PbBr₆ byproduct in any of the syntheses.

Action: We have included new data; please see our response to R3Q4.

✓R4Q2: Water have been shown to promote the formation of NRs but is not discussed here (for instance, “Efficient Interfacial Synthesis Strategy for Perovskite CsPbBr₃ Nanorods in the Biphasic Solution”, Adv. Mater. Technol. 2022, 7, 2200131, DOI: 10.1002/admt.202200131).

Response: We appreciate putting our attention to this approach, which uses a macroscopic water interface. This interesting synthesis scheme is indeed very different from ours. The water content in our synthesis is far lower than in the publication mentioned by the Reviewer (0.2% versus 5%). Due to the lack of water in our synthesis, ions are not soluble in the nonpolar solvent (toluene). In turn, micelles and nanoclusters with ligand shells dominate our synthesis.

Action: We added *“Importantly, freely dissolved precursor ions in the solution are not expected to play a significant role due to their insolubility in toluene. This distinguishes the reaction pathway from, e.g., biphasic syntheses where water can serve as an ion reservoir⁴⁷.”* (page 5) and the citation.

✓R4Q3: The authors don’t address control over the aspect ratio of the NRs.

Response: Indeed, the lateral size of the platelets and rod length was not our main concern. Instead, we focus on the thinnest dimension, which dictates quantum confinement. However, we agree that aspect ratio is interesting on its own. We obtain aspect ratios of up to (21.5±6.7) for the longest nanorods by increasing the concentration of the injected Cs-oleate precursor.

Action: **Changes to the manuscript:** We have included additional TEM images, size distributions and ex situ optical characterisations in Supplementary Fig. 32 and Supplementary Fig. 33 to demonstrate the tunability of nanorod aspect ratio as well as nanoplatelet lateral size in our synthesis approach.

✓R4Q4: The authors may add a reference: “Shape-controlled synthesis of CsPbBr₃ nanorods with bright pure blue emission and high stability”, J. Mater. Chem. C, 2024,12, 4234-4242, DOI:10.1039/D3TC04681H.

Response: We thank the Reviewer for making us aware of this publication.

Action: We have added this reference to the introduction (reference¹²).

REVIEWER 3 ATTACHMENT:

This is a straightforward manuscript giving insight into the growth of anisotropic CsPbBr₃ nanorods and platelets using *in situ* X-ray and spectroscopy techniques. I think the manuscript can be improved by providing more clarity on several points, discussed below, and I also think more citations and a stronger connection to comparable research can be provided. Once these issues are addressed, I think the manuscript would be ready for publication.

- Page 3, Line 63: There are studies that demonstrate slow reaction kinetics, and maintain the cuboid shape of a conventional fast synthesis. I encourage the authors to read and cite Chem. Mater. 2019, 31, 20, 8551–8557

-Page 3, lines 80 and 83: What conditions? The wording here is a little confusing as it is left a little vague.

-Page four, figure 1. Nanorods are shown to have dimensions of 1.8 nm (width) and 15 nm lengths. The emission peak for these CsPbBr₃ nanorods is 460 nm. The emission seems much more red-shifted than I believe has been reported in other studies of quantum-confined nanorods. It would be helpful to compare these samples with the other examples in the literature, especially e.g. *Nano Lett.* 2022, 22, 20, 8355–8362 and *ACS Nano* 2022, 16, 5, 8318–8328 – I wonder if there is a systematic difference in how the authors of these different manuscripts are determining rod width.

Relatedly, the authors perform SAXS and WAXS, but do not show powder XRD patterns. Are the samples produced pure perovskite? Commonly, lead-deficient Cs₄PbBr₆ is produced during a synthesis, and may be expected to vary as a side-product across the parameter space being studied. Can the authors correlate the XRD pattern of the product with the lattice planes that are resolved in the TEM images of nanorods and nanoplatelets?

Additionally, can the authors identify which facet these rods “grown” along? Are all the nanorods oriented in the direction of a specific facet? This would require looking at the lattice planes on high resolution TEM images.

-Page 4 line 94 “The synthesis is conducted at room temperature in ambient conditions”, can the authors be more specific about “ambient conditions”? If it is referring to temperature, then it was already specified. In the case where ambient conditions in reference to the ambient atmosphere (i.e. oxygen, moisture etc.), there seems to be a discrepancy between the custom-built reaction cell being filled with Helium and it being under “ambient conditions”, could the authors please be more specific?

- Page 4, Line 100: It is not clear how the authors can deduce the role of Br versus Pb for determining product - these ions are always present in the same stoichiometric ratio by the choice of precursor (PbBr₂). Other work has identified that Pb plays little role in the synthesis (see Chem. Mater. 2019, 31, 20, 8551–8557). Perhaps the authors have done control studies that identify the Cs/Pb ratio, uniquely compared to the Cs/Br ratio?

- Page 5, Line 140: How many units of PbBr_2 does the “precursor micelle” correspond to? Is it a crystalline cluster? This is a very interesting result. Is this reported in other works, and the authors are corroborating that finding? It would be very good to say one way or the other, and cite appropriately if so. What is the stoichiometry of that micelle? It seems quite important to understand if the precursor micelle is the template for the mesoscale hexagonal lattice that forms later. I think more clarity/discussion about this species, and the exact transformation into the next proposed stage of the growth would be very helpful for the manuscript. Please be clear about what is proven versus what is hypothesized but unsubstantiated.

- Page 4, panel C: Why is the I_{001} signal not monotonic?